# Semi-supervised Knowledge Transfer for Deep Learning from Private Training Data

**Nicolas Papernot**[*]
Pennsylvania State University
ngp5056@cse.psu.edu

**Martín Abadi**
Google Brain
abadi@google.com

**Úlfar Erlingsson**
Google
ulfar@google.com

**Ian Goodfellow**
Google Brain[†]
goodfellow@google.com

**Kunal Talwar**
Google Brain
kunal@google.com

## Abstract

Some machine learning applications involve training data that is sensitive, such as the medical histories of patients in a clinical trial. A model may inadvertently and implicitly store some of its training data; careful analysis of the model may therefore reveal sensitive information.

To address this problem, we demonstrate a generally applicable approach to providing strong privacy guarantees for training data: *Private Aggregation of Teacher Ensembles* (PATE). The approach combines, in a black-box fashion, multiple models trained with disjoint datasets, such as records from different subsets of users. Because they rely directly on sensitive data, these models are not published, but instead used as "teachers" for a "student" model. The student learns to predict an output chosen by noisy voting among all of the teachers, and cannot directly access an individual teacher or the underlying data or parameters. The student's privacy properties can be understood both intuitively (since no single teacher and thus no single dataset dictates the student's training) and formally, in terms of differential privacy. These properties hold even if an adversary can not only query the student but also inspect its internal workings.

Compared with previous work, the approach imposes only weak assumptions on how teachers are trained: it applies to any model, including non-convex models like DNNs. We achieve state-of-the-art privacy/utility trade-offs on MNIST and SVHN thanks to an improved privacy analysis and semi-supervised learning.

## 1 Introduction

Some machine learning applications with great benefits are enabled only through the analysis of sensitive data, such as users' personal contacts, private photographs or correspondence, or even medical records or genetic sequences (Alipanahi et al., 2015; Kannan et al., 2016; Kononenko, 2001; Sweeney, 1997). Ideally, in those cases, the learning algorithms would protect the privacy of users' training data, e.g., by guaranteeing that the output model generalizes away from the specifics of any individual user. Unfortunately, established machine learning algorithms make no such guarantee; indeed, though state-of-the-art algorithms generalize well to the test set, they continue to overfit on specific training examples in the sense that some of these examples are implicitly memorized.

Recent attacks exploiting this implicit memorization in machine learning have demonstrated that private, sensitive training data can be recovered from models. Such attacks can proceed directly, by analyzing internal model parameters, but also indirectly, by repeatedly querying opaque models to gather data for the attack's analysis. For example, Fredrikson et al. (2015) used hill-climbing on the output probabilities of a computer-vision classifier to reveal individual faces from the training data.

---

[*]Work done while the author was at Google.
[†]Work done both at Google Brain and at OpenAI.

Because of those demonstrations—and because privacy guarantees must apply to worst-case outliers, not only the average—any strategy for protecting the privacy of training data should prudently assume that attackers have unfettered access to internal model parameters.

To protect the privacy of training data, this paper improves upon a specific, structured application of the techniques of knowledge aggregation and transfer (Breiman, 1994), previously explored by Nissim et al. (2007), Pathak et al. (2010), and particularly Hamm et al. (2016). In this strategy, first, an ensemble (Dietterich, 2000) of teacher models is trained on disjoint subsets of the sensitive data. Then, using auxiliary, unlabeled non-sensitive data, a student model is trained on the aggregate output of the ensemble, such that the student learns to accurately mimic the ensemble. Intuitively, this strategy ensures that the student does not depend on the details of any single sensitive training data point (e.g., of any single user), and, thereby, the privacy of the training data is protected even if attackers can observe the student's internal model parameters.

This paper shows how this strategy's privacy guarantees can be strengthened by restricting student training to a limited number of teacher votes, and by revealing only the topmost vote after carefully adding random noise. We call this strengthened strategy PATE, for *Private Aggregation of Teacher Ensembles*. Furthermore, we introduce an improved privacy analysis that makes the strategy generally applicable to machine learning algorithms with high utility and meaningful privacy guarantees—in particular, when combined with semi-supervised learning.

To establish strong privacy guarantees, it is important to limit the student's access to its teachers, so that the student's exposure to teachers' knowledge can be meaningfully quantified and bounded. Fortunately, there are many techniques for speeding up knowledge transfer that can reduce the rate of student/teacher consultation during learning. We describe several techniques in this paper, the most effective of which makes use of generative adversarial networks (GANs) (Goodfellow et al., 2014) applied to semi-supervised learning, using the implementation proposed by Salimans et al. (2016). For clarity, we use the term PATE-G when our approach is combined with generative, semi-supervised methods. Like all semi-supervised learning methods, PATE-G assumes the student has access to additional, unlabeled data, which, in this context, must be public or non-sensitive. This assumption should not greatly restrict our method's applicability: even when learning on sensitive data, a non-overlapping, unlabeled set of data often exists, from which semi-supervised methods can extract distribution priors. For instance, public datasets exist for text and images, and for medical data.

It seems intuitive, or even obvious, that a student machine learning model will provide good privacy when trained without access to sensitive training data, apart from a few, noisy votes from a teacher quorum. However, intuition is not sufficient because privacy properties can be surprisingly hard to reason about; for example, even a single data item can greatly impact machine learning models trained on a large corpus (Chaudhuri et al., 2011). Therefore, to limit the effect of any single sensitive data item on the student's learning, precisely and formally, we apply the well-established, rigorous standard of differential privacy (Dwork & Roth, 2014). Like all differentially private algorithms, our learning strategy carefully adds noise, so that the privacy impact of each data item can be analyzed and bounded. In particular, we dynamically analyze the sensitivity of the teachers' noisy votes; for this purpose, we use the state-of-the-art moments accountant technique from Abadi et al. (2016), which tightens the privacy bound when the topmost vote has a large quorum. As a result, for MNIST and similar benchmark learning tasks, our methods allow students to provide excellent utility, while our analysis provides meaningful worst-case guarantees. In particular, we can bound the metric for privacy loss (the differential-privacy $\varepsilon$) to a range similar to that of existing, real-world privacy-protection mechanisms, such as Google's RAPPOR (Erlingsson et al., 2014).

Finally, it is an important advantage that our learning strategy and our privacy analysis do not depend on the details of the machine learning techniques used to train either the teachers or their student. Therefore, the techniques in this paper apply equally well for deep learning methods, or any such learning methods with large numbers of parameters, as they do for shallow, simple techniques. In comparison, Hamm et al. (2016) guarantee privacy only conditionally, for a restricted class of student classifiers—in effect, limiting applicability to logistic regression with convex loss. Also, unlike the methods of Abadi et al. (2016), which represent the state-of-the-art in differentially-private deep learning, our techniques make no assumptions about details such as batch selection, the loss function, or the choice of the optimization algorithm. Even so, as we show in experiments on

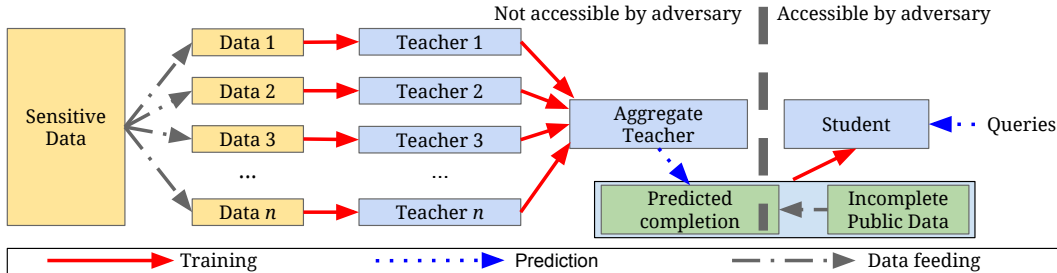

Figure 1: Overview of the approach: (1) an ensemble of teachers is trained on disjoint subsets of the sensitive data, (2) a student model is trained on public data labeled using the ensemble.

MNIST and SVHN, our techniques provide a privacy/utility tradeoff that equals or improves upon bespoke learning methods such as those of Abadi et al. (2016).

Section 5 further discusses the related work. Building on this related work, our contributions are as follows:

- We demonstrate a general machine learning strategy, the PATE approach, that provides differential privacy for training data in a "black-box" manner, i.e., independent of the learning algorithm, as demonstrated by Section 4 and Appendix C.

- We improve upon the strategy outlined in Hamm et al. (2016) for learning machine models that protect training data privacy. In particular, our student only accesses the teachers' top vote and the model does not need to be trained with a restricted class of convex losses.

- We explore four different approaches for reducing the student's dependence on its teachers, and show how the application of GANs to semi-supervised learning of Salimans et al. (2016) can greatly reduce the privacy loss by radically reducing the need for supervision.

- We present a new application of the moments accountant technique from Abadi et al. (2016) for improving the differential-privacy analysis of knowledge transfer, which allows the training of students with meaningful privacy bounds.

- We evaluate our framework on MNIST and SVHN, allowing for a comparison of our results with previous differentially private machine learning methods. Our classifiers achieve an $(\varepsilon, \delta)$ differential-privacy bound of $(2.04, 10^{-5})$ for MNIST and $(8.19, 10^{-6})$ for SVHN, respectively with accuracy of $98.00\%$ and $90.66\%$. In comparison, for MNIST, Abadi et al. (2016) obtain a looser $(8, 10^{-5})$ privacy bound and $97\%$ accuracy. For SVHN, Shokri & Shmatikov (2015) report approx. $92\%$ accuracy with $\varepsilon > 2$ per each of 300,000 model parameters, naively making the total $\varepsilon > 600,000$, which guarantees no meaningful privacy.

- Finally, we show that the PATE approach can be successfully applied to other model structures and to datasets with different characteristics. In particular, in Appendix C PATE protects the privacy of medical data used to train a model based on random forests.

Our results are encouraging, and highlight the benefits of combining a learning strategy based on semi-supervised knowledge transfer with a precise, data-dependent privacy analysis. However, the most appealing aspect of this work is probably that its guarantees can be compelling to both an expert and a non-expert audience. In combination, our techniques simultaneously provide both an intuitive and a rigorous guarantee of training data privacy, without sacrificing the utility of the targeted model. This gives hope that users will increasingly be able to confidently and safely benefit from machine learning models built from their sensitive data.

## 2 PRIVATE LEARNING WITH ENSEMBLES OF TEACHERS

In this section, we introduce the specifics of the PATE approach, which is illustrated in Figure 1. We describe how the data is partitioned to train an ensemble of teachers, and how the predictions made by this ensemble are noisily aggregated. In addition, we discuss how GANs can be used in training the student, and distinguish PATE-G variants that improve our approach using generative, semi-supervised methods.

## 2.1 TRAINING THE ENSEMBLE OF TEACHERS

**Data partitioning and teachers:** Instead of training a single model to solve the task associated with dataset $(X, Y)$, where $X$ denotes the set of inputs, and $Y$ the set of labels, we partition the data in $n$ disjoint sets $(X_n, Y_n)$ and train a model separately on each set. As evaluated in Section 4.1, assuming that $n$ is not too large with respect to the dataset size and task complexity, we obtain $n$ classifiers $f_i$ called teachers. We then deploy them as an ensemble making predictions on unseen inputs $x$ by querying each teacher for a prediction $f_i(x)$ and aggregating these into a single prediction.

**Aggregation:** The privacy guarantees of this teacher ensemble stems from its aggregation. Let $m$ be the number of classes in our task. The label count for a given class $j \in [m]$ and an input $\vec{x}$ is the number of teachers that assigned class $j$ to input $\vec{x}$: $n_j(\vec{x}) = |\{i : i \in [n], f_i(\vec{x}) = j\}|$. If we simply apply *plurality*—use the label with the largest count—the ensemble's decision may depend on a single teacher's vote. Indeed, when two labels have a vote count differing by at most one, there is a tie: the aggregated output changes if one teacher makes a different prediction. We add random noise to the vote counts $n_j$ to introduce ambiguity:

$$f(x) = \arg\max_j \left\{ n_j(\vec{x}) + Lap\left(\frac{1}{\gamma}\right) \right\} \tag{1}$$

In this equation, $\gamma$ is a privacy parameter and $Lap(b)$ the Laplacian distribution with location 0 and scale $b$. The parameter $\gamma$ influences the privacy guarantee we can prove. Intuitively, a large $\gamma$ leads to a strong privacy guarantee, but can degrade the accuracy of the labels, as the noisy maximum $f$ above can differ from the true plurality.

While we could use an $f$ such as above to make predictions, the noise required would increase as we make more predictions, making the model useless after a bounded number of queries. Furthermore, privacy guarantees do not hold when an adversary has access to the model parameters. Indeed, as each teacher $f_i$ was trained without taking into account privacy, it is conceivable that they have sufficient capacity to retain details of the training data. To address these limitations, we train another model, the student, using a fixed number of labels predicted by the teacher ensemble.

## 2.2 SEMI-SUPERVISED TRANSFER OF THE KNOWLEDGE FROM AN ENSEMBLE TO A STUDENT

We train a student on nonsensitive and unlabeled data, some of which we label using the aggregation mechanism. This student model is the one deployed, in lieu of the teacher ensemble, so as to fix the privacy loss to a value that does not grow with the number of user queries made to the student model. Indeed, the privacy loss is now determined by the number of queries made to the teacher ensemble during student training and does not increase as end-users query the deployed student model. Thus, the privacy of users who contributed to the original training dataset is preserved even if the student's architecture and parameters are public or reverse-engineered by an adversary.

We considered several techniques to trade-off the student model's quality with the number of labels it needs to access: distillation, active learning, semi-supervised learning (see Appendix B). Here, we only describe the most successful one, used in PATE-G: semi-supervised learning with GANs.

**Training the student with GANs:** The GAN framework involves two machine learning models, a *generator* and a *discriminator*. They are trained in a competing fashion, in what can be viewed as a two-player game (Goodfellow et al., 2014). The generator produces samples from the data distribution by transforming vectors sampled from a Gaussian distribution. The discriminator is trained to distinguish samples artificially produced by the generator from samples part of the real data distribution. Models are trained via simultaneous gradient descent steps on both players' costs. In practice, these dynamics are often difficult to control when the strategy set is non-convex (e.g., a DNN). In their application of GANs to semi-supervised learning, Salimans et al. (2016) made the following modifications. The discriminator is extended from a binary classifier (data vs. generator sample) to a multi-class classifier (one of $k$ classes of data samples, plus a class for generated samples). This classifier is then trained to classify labeled real samples in the correct class, unlabeled real samples in any of the $k$ classes, and the generated samples in the additional class.

Although no formal results currently explain why yet, the technique was empirically demonstrated to greatly improve semi-supervised learning of classifiers on several datasets, especially when the classifier is trained with *feature matching* loss (Salimans et al., 2016).

Training the student in a semi-supervised fashion makes better use of the entire data available to the student, while still only labeling a subset of it. Unlabeled inputs are used in unsupervised learning to estimate a good prior for the distribution. Labeled inputs are then used for supervised learning.

## 3 PRIVACY ANALYSIS OF THE APPROACH

We now analyze the differential privacy guarantees of our PATE approach. Namely, we keep track of the privacy budget throughout the student's training using the moments accountant (Abadi et al., 2016). When teachers reach a strong quorum, this allows us to bound privacy costs more strictly.

### 3.1 DIFFERENTIAL PRIVACY PRELIMINARIES AND A SIMPLE ANALYSIS OF PATE

Differential privacy (Dwork et al., 2006b; Dwork, 2011) has established itself as a strong standard. It provides privacy guarantees for algorithms analyzing databases, which in our case is a machine learning training algorithm processing a training dataset. Differential privacy is defined using pairs of adjacent databases: in the present work, these are datasets that only differ by one training example. Recall the following variant of differential privacy introduced in Dwork et al. (2006a).

**Definition 1.** *A randomized mechanism $\mathcal{M}$ with domain $\mathcal{D}$ and range $\mathcal{R}$ satisfies $(\varepsilon, \delta)$-differential privacy if for any two adjacent inputs $d, d' \in \mathcal{D}$ and for any subset of outputs $S \subseteq \mathcal{R}$ it holds that:*

$$\Pr[\mathcal{M}(d) \in S] \le e^{\varepsilon} \Pr[\mathcal{M}(d') \in S] + \delta. \tag{2}$$

It will be useful to define the *privacy loss* and the *privacy loss random variable*. They capture the differences in the probability distribution resulting from running $\mathcal{M}$ on $d$ and $d'$.

**Definition 2.** *Let $\mathcal{M}: \mathcal{D} \to \mathcal{R}$ be a randomized mechanism and $d, d'$ a pair of adjacent databases. Let aux denote an auxiliary input. For an outcome $o \in \mathcal{R}$, the privacy loss at $o$ is defined as:*

$$c(o; \mathcal{M}, \textit{aux}, d, d') \triangleq \log \frac{\Pr[\mathcal{M}(\textit{aux}, d) = o]}{\Pr[\mathcal{M}(\textit{aux}, d') = o]}. \tag{3}$$

*The privacy loss random variable $C(\mathcal{M}, \textit{aux}, d, d')$ is defined as $c(\mathcal{M}(d); \mathcal{M}, \textit{aux}, d, d')$, i.e. the random variable defined by evaluating the privacy loss at an outcome sampled from $\mathcal{M}(d)$.*

A natural way to bound our approach's privacy loss is to first bound the privacy cost of each label queried by the student, and then use the strong composition theorem (Dwork et al., 2010) to derive the total cost of training the student. For neighboring databases $d, d'$, each teacher gets the same training data partition (that is, the same for the teacher with $d$ and with $d'$, not the same across teachers), with the exception of one teacher whose corresponding training data partition differs. Therefore, the label counts $n_j(\vec{x})$ for any example $\vec{x}$, on $d$ and $d'$ differ by at most 1 in at most two locations. In the next subsection, we show that this yields loose guarantees.

### 3.2 THE MOMENTS ACCOUNTANT: A BUILDING BLOCK FOR BETTER ANALYSIS

To better keep track of the privacy cost, we use recent advances in privacy cost accounting. The moments accountant was introduced by Abadi et al. (2016), building on previous work (Bun & Steinke, 2016; Dwork & Rothblum, 2016; Mironov, 2016).

**Definition 3.** *Let $\mathcal{M}: \mathcal{D} \to \mathcal{R}$ be a randomized mechanism and $d, d'$ a pair of adjacent databases. Let aux denote an auxiliary input. The moments accountant is defined as:*

$$\alpha_{\mathcal{M}}(\lambda) \triangleq \max_{\textit{aux}, d, d'} \alpha_{\mathcal{M}}(\lambda; \textit{aux}, d, d') \tag{4}$$

*where $\alpha_{\mathcal{M}}(\lambda; \textit{aux}, d, d') \triangleq \log \mathbb{E}[\exp(\lambda C(\mathcal{M}, \textit{aux}, d, d'))]$ is the moment generating function of the privacy loss random variable.*

The following properties of the moments accountant are proved in Abadi et al. (2016).

**Theorem 1.** *1. [Composability] Suppose that a mechanism $\mathcal{M}$ consists of a sequence of adaptive mechanisms $\mathcal{M}_1, \ldots, \mathcal{M}_k$ where $\mathcal{M}_i \colon \prod_{j=1}^{i-1} \mathcal{R}_j \times \mathcal{D} \to \mathcal{R}_i$. Then, for any output sequence $o_1, \ldots, o_{k-1}$ and any $\lambda$*

$$\alpha_{\mathcal{M}}(\lambda; d, d') = \sum_{i=1}^{k} \alpha_{\mathcal{M}_i}(\lambda; o_1, \ldots, o_{i-1}, d, d'),$$

*where $\alpha_{\mathcal{M}}$ is conditioned on $\mathcal{M}_i$'s output being $o_i$ for $i < k$.*

*2. [Tail bound] For any $\varepsilon > 0$, the mechanism $\mathcal{M}$ is $(\varepsilon, \delta)$-differentially private for*

$$\delta = \min_{\lambda} \exp(\alpha_{\mathcal{M}}(\lambda) - \lambda\varepsilon).$$

We write down two important properties of the aggregation mechanism from Section 2. The first property is proved in Dwork & Roth (2014), and the second follows from Bun & Steinke (2016).

**Theorem 2.** *Suppose that on neighboring databases $d, d'$, the label counts $n_j$ differ by at most 1 in each coordinate. Let $\mathcal{M}$ be the mechanism that reports $\arg\max_j \left\{ n_j + Lap(\frac{1}{\gamma}) \right\}$. Then $\mathcal{M}$ satisfies $(2\gamma, 0)$-differential privacy. Moreover, for any $l$, aux, $d$ and $d'$,*

$$\alpha(l; \textit{aux}, d, d') \leq 2\gamma^2 l(l+1) \tag{5}$$

At each step, we use the aggregation mechanism with noise $Lap(\frac{1}{\gamma})$ which is $(2\gamma, 0)$-DP. Thus over $T$ steps, we get $(4T\gamma^2 + 2\gamma\sqrt{2T \ln \frac{1}{\delta}}, \delta)$-differential privacy. This can be rather large: plugging in values that correspond to our SVHN result, $\gamma = 0.05, T = 1000, \delta = 1\mathrm{e}{-6}$ gives us $\varepsilon \approx 26$ or alternatively plugging in values that correspond to our MNIST result, $\gamma = 0.05, T = 100, \delta = 1\mathrm{e}{-5}$ gives us $\varepsilon \approx 5.80$.

### 3.3 A PRECISE, DATA-DEPENDENT PRIVACY ANALYSIS OF PATE

Our data-dependent privacy analysis takes advantage of the fact that when the quorum among the teachers is very strong, the majority outcome has overwhelming likelihood, in which case the privacy cost is small whenever this outcome occurs. The moments accountant allows us analyze the composition of such mechanisms in a unified framework.

The following theorem, proved in Appendix A, provides a data-dependent bound on the moments of any differentially private mechanism where some specific outcome is very likely.

**Theorem 3.** *Let $\mathcal{M}$ be $(2\gamma, 0)$-differentially private and $q \geq \Pr[\mathcal{M}(d) \neq o^*]$ for some outcome $o^*$. Let $l, \gamma \geq 0$ and $q < \frac{e^{2\gamma} - 1}{e^{4\gamma} - 1}$. Then for any aux and any neighbor $d'$ of $d$, $\mathcal{M}$ satisfies*

$$\alpha(l; \textit{aux}, d, d') \leq \log((1-q)\Big(\frac{1-q}{1-e^{2\gamma}q}\Big)^l + q\exp(2\gamma l)).$$

To upper bound $q$ for our aggregation mechanism, we use the following simple lemma, also proved in Appendix A.

**Lemma 4.** *Let $\mathbf{n}$ be the label score vector for a database $d$ with $n_{j^*} \geq n_j$ for all $j$. Then*

$$\Pr[\mathcal{M}(d) \neq j^*] \leq \sum_{j \neq j^*} \frac{2 + \gamma(n_{j^*} - n_j)}{4\exp(\gamma(n_{j^*} - n_j))}$$

This allows us to upper bound $q$ for a specific score vector $\mathbf{n}$, and hence bound specific moments. We take the smaller of the bounds we get from Theorems 2 and 3. We compute these moments for a few values of $\lambda$ (integers up to 8). Theorem 1 allows us to add these bounds over successive steps, and derive an $(\varepsilon, \delta)$ guarantee from the final $\alpha$. Interested readers are referred to the script that we used to empirically compute these bounds, which is released along with our code: `https://github.com/tensorflow/models/tree/master/differential_privacy/multiple_teachers`

Since the privacy moments are themselves now data dependent, the final $\varepsilon$ is itself data-dependent and should not be revealed. To get around this, we bound the *smooth sensitivity* (Nissim et al., 2007) of the moments and add noise proportional to it to the moments themselves. This gives us a differentially private estimate of the privacy cost. Our evaluation in Section 4 ignores this overhead and reports the un-noised values of $\varepsilon$. Indeed, in our experiments on MNIST and SVHN, the scale of the noise one needs to add to the released $\varepsilon$ is smaller than 0.5 and 1.0 respectively.

How does the number of teachers affect the privacy cost? Recall that the student uses a noisy label computed in (1) which has a parameter $\gamma$. To ensure that the noisy label is likely to be the correct one, the noise scale $\frac{1}{\gamma}$ should be small compared to the the additive gap between the two largest vales of $n_j$. While the exact dependence of $\gamma$ on the privacy cost in Theorem 3 is subtle, as a general principle, a smaller $\gamma$ leads to a smaller privacy cost. Thus, a larger gap translates to a smaller privacy cost. Since the gap itself increases with the number of teachers, having more teachers would lower the privacy cost. This is true up to a point. With $n$ teachers, each teacher only trains on a $\frac{1}{n}$ fraction of the training data. For large enough $n$, each teachers will have too little training data to be accurate.

To conclude, we note that our analysis is rather conservative in that it pessimistically assumes that, even if just one example in the training set for one teacher changes, the classifier produced by that teacher may change arbitrarily. One advantage of our approach, which enables its wide applicability, is that our analysis does not require any assumptions about the workings of the teachers. Nevertheless, we expect that stronger privacy guarantees may perhaps be established in specific settings—when assumptions can be made on the learning algorithm used to train the teachers.

## 4 EVALUATION

In our evaluation of PATE and its generative variant PATE-G, we first train a teacher ensemble for each dataset. The trade-off between the accuracy and privacy of labels predicted by the ensemble is greatly dependent on the number of teachers in the ensemble: being able to train a large set of teachers is essential to support the injection of noise yielding strong privacy guarantees while having a limited impact on accuracy. Second, we minimize the privacy budget spent on learning the student by training it with as few queries to the ensemble as possible.

Our experiments use MNIST and the extended SVHN datasets. Our MNIST model stacks two convolutional layers with max-pooling and one fully connected layer with ReLUs. When trained on the entire dataset, the non-private model has a $99.18\%$ test accuracy. For SVHN, we add two hidden layers.[1] The non-private model achieves a $92.8\%$ test accuracy, which is shy of the state-of-the-art. However, we are primarily interested in comparing the private student's accuracy with the one of a non-private model trained on the entire dataset, for different privacy guarantees. The source code for reproducing the results in this section is available on GitHub.[2]

### 4.1 TRAINING AN ENSEMBLE OF TEACHERS PRODUCING PRIVATE LABELS

As mentioned above, compensating the noise introduced by the Laplacian mechanism presented in Equation 1 requires large ensembles. We evaluate the extent to which the two datasets considered can be partitioned with a reasonable impact on the performance of individual teachers. Specifically, we show that for MNIST and SVHN, we are able to train ensembles of 250 teachers. Their aggregated predictions are accurate despite the injection of large amounts of random noise to ensure privacy. The aggregation mechanism output has an accuracy of $93.18\%$ for MNIST and $87.79\%$ for SVHN, when evaluated on their respective test sets, while each query has a low privacy budget of $\varepsilon = 0.05$.

**Prediction accuracy:** All other things being equal, the number $n$ of teachers is limited by a trade-off between the classification task's complexity and the available data. We train $n$ teachers by partitioning the training data $n$-way. Larger values of $n$ lead to larger absolute gaps, hence potentially allowing for a larger noise level and stronger privacy guarantees. At the same time, a larger $n$ implies a smaller training dataset for each teacher, potentially reducing the teacher accuracy. We empirically find appropriate values of $n$ for the MNIST and SVHN datasets by measuring the test

---

[1] The model is adapted from `https://www.tensorflow.org/tutorials/deep_cnn`

[2] `https://github.com/tensorflow/models/tree/master/differential_privacy/multiple_teachers`

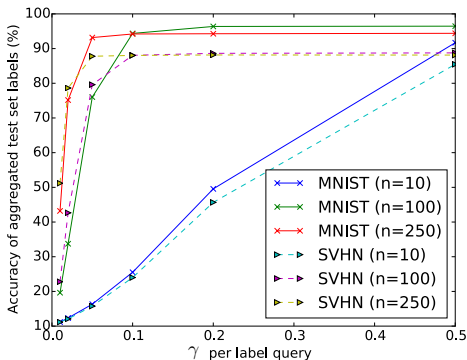 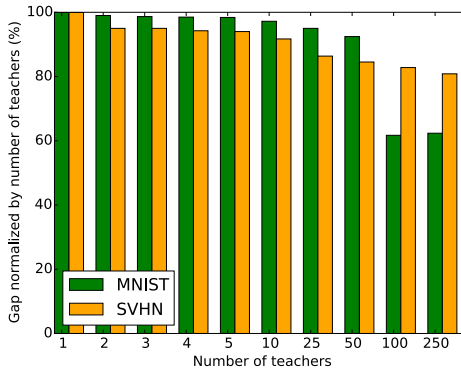

Figure 2: **How much noise can be injected to a query?** Accuracy of the noisy aggregation for three MNIST and SVHN teacher ensembles and varying $\gamma$ value per query. The noise introduced to achieve a given $\gamma$ scales inversely proportionally to the value of $\gamma$: small values of $\gamma$ on the left of the axis correspond to large noise amplitudes and large $\gamma$ values on the right to small noise.

Figure 3: **How certain is the aggregation of teacher predictions?** Gap between the number of votes assigned to the most and second most frequent labels normalized by the number of teachers in an ensemble. Larger gaps indicate that the ensemble is confident in assigning the labels, and will be robust to more noise injection. Gaps were computed by averaging over the test data.

set accuracy of each teacher trained on one of the $n$ partitions of the training data. We find that even for $n = 250$, the average test accuracy of individual teachers is $83.86\%$ for MNIST and $83.18\%$ for SVHN. The larger size of SVHN compensates its increased task complexity.

**Prediction confidence:** As outlined in Section 3, the privacy of predictions made by an ensemble of teachers intuitively requires that a quorum of teachers generalizing well agree on identical labels. This observation is reflected by our data-dependent privacy analysis, which provides stricter privacy bounds when the quorum is strong. We study the disparity of labels assigned by teachers. In other words, we count the number of votes for each possible label, and measure the difference in votes between the most popular label and the second most popular label, i.e., the *gap*. If the gap is small, introducing noise during aggregation might change the label assigned from the first to the second. Figure 3 shows the gap normalized by the total number of teachers $n$. As $n$ increases, the gap remains larger than $60\%$ of the teachers, allowing for aggregation mechanisms to output the correct label in the presence of noise.

**Noisy aggregation:** For MNIST and SVHN, we consider three ensembles of teachers with varying number of teachers $n \in \{10, 100, 250\}$. For each of them, we perturb the vote counts with Laplacian noise of inversed scale $\gamma$ ranging between $0.01$ and $1$. This choice is justified below in Section 4.2. We report in Figure 2 the accuracy of test set labels inferred by the noisy aggregation mechanism for these values of $\varepsilon$. Notice that the number of teachers needs to be large to compensate for the impact of noise injection on the accuracy.

## 4.2 SEMI-SUPERVISED TRAINING OF THE STUDENT WITH PRIVACY

The noisy aggregation mechanism labels the student's unlabeled training set in a privacy-preserving fashion. To reduce the privacy budget spent on student training, we are interested in making as few label queries to the teachers as possible. We therefore use the semi-supervised training approach described previously. Our MNIST and SVHN students with $(\varepsilon, \delta)$ differential privacy of $(2.04, 10^{-5})$ and $(8.19, 10^{-6})$ achieve accuracies of $98.00\%$ and $90.66\%$. These results improve the differential privacy state-of-the-art for these datasets. Abadi et al. (2016) previously obtained $97\%$ accuracy with a $(8, 10^{-5})$ bound on MNIST, starting from an inferior baseline model without privacy. Shokri & Shmatikov (2015) reported about $92\%$ accuracy on SVHN with $\varepsilon > 2$ per model parameter and a model with over 300,000 parameters. Naively, this corresponds to a total $\varepsilon > 600,000$.

| Dataset | $\varepsilon$ | $\delta$ | Queries | Non-Private Baseline | Student Accuracy |
|---------|------|-----------|---------|---------------------|------------------|
| MNIST | 2.04 | $10^{-5}$ | 100 | 99.18% | 98.00% |
| MNIST | 8.03 | $10^{-5}$ | 1000 | 99.18% | 98.10% |
| SVHN | 5.04 | $10^{-6}$ | 500 | 92.80% | 82.72% |
| SVHN | 8.19 | $10^{-6}$ | 1000 | 92.80% | 90.66% |

Figure 4: **Utility and privacy of the semi-supervised students:** each row is a variant of the student model trained with generative adversarial networks in a semi-supervised way, with a different number of label queries made to the teachers through the noisy aggregation mechanism. The last column reports the accuracy of the student and the second and third column the bound $\varepsilon$ and failure probability $\delta$ of the $(\varepsilon, \delta)$ differential privacy guarantee.

We apply semi-supervised learning with GANs to our problem using the following setup for each dataset. In the case of MNIST, the student has access to 9,000 samples, among which a subset of either 100, 500, or 1,000 samples are labeled using the noisy aggregation mechanism discussed in Section 2.1. Its performance is evaluated on the 1,000 remaining samples of the test set. Note that this may increase the variance of our test set accuracy measurements, when compared to those computed over the entire test data. For the MNIST dataset, we randomly shuffle the test set to ensure that the different classes are balanced when selecting the (small) subset labeled to train the student. For SVHN, the student has access to 10,000 training inputs, among which it labels 500 or 1,000 samples using the noisy aggregation mechanism. Its performance is evaluated on the remaining 16,032 samples. For both datasets, the ensemble is made up of 250 teachers. We use Laplacian scale of 20 to guarantee an individual query privacy bound of $\varepsilon = 0.05$. These parameter choices are motivated by the results from Section 4.1.

In Figure 4, we report the values of the $(\varepsilon, \delta)$ differential privacy guarantees provided and the corresponding student accuracy, as well as the number of queries made by each student. The MNIST student is able to learn a 98% accurate model, which is shy of 1% when compared to the accuracy of a model learned with the entire training set, with only 100 label queries. This results in a strict differentially private bound of $\varepsilon = 2.04$ for a failure probability fixed at $10^{-5}$. The SVHN student achieves 90.66% accuracy, which is also comparable to the 92.80% accuracy of one teacher learned with the entire training set. The corresponding privacy bound is $\varepsilon = 8.19$, which is higher than for the MNIST dataset, likely because of the larger number of queries made to the aggregation mechanism.

We observe that our private student outperforms the aggregation's output in terms of accuracy, with or without the injection of Laplacian noise. While this shows the power of semi-supervised learning, the student may not learn as well on different kinds of data (e.g., medical data), where categories are not explicitly designed by humans to be salient in the input space. Encouragingly, as Appendix C illustrates, the PATE approach can be successfully applied to at least some examples of such data.

## 5 DISCUSSION AND RELATED WORK

Several privacy definitions are found in the literature. For instance, *k-anonymity* requires information about an individual to be indistinguishable from at least $k - 1$ other individuals in the dataset (L. Sweeney, 2002). However, its lack of randomization gives rise to caveats (Dwork & Roth, 2014), and attackers can infer properties of the dataset (Aggarwal, 2005). An alternative definition, *differential privacy*, established itself as a rigorous standard for providing privacy guarantees (Dwork et al., 2006b). In contrast to $k$-anonymity, differential privacy is a property of the randomized algorithm and not the dataset itself.

A variety of approaches and mechanisms can guarantee differential privacy. Erlingsson et al. (2014) showed that randomized response, introduced by Warner (1965), can protect crowd-sourced data collected from software users to compute statistics about user behaviors. Attempts to provide differential privacy for machine learning models led to a series of efforts on shallow machine learning models, including work by Bassily et al. (2014); Chaudhuri & Monteleoni (2009); Pathak et al. (2011); Song et al. (2013), and Wainwright et al. (2012).

A privacy-preserving distributed SGD algorithm was introduced by Shokri & Shmatikov (2015). It applies to non-convex models. However, its privacy bounds are given per-parameter, and the large number of parameters prevents the technique from providing a meaningful privacy guarantee. Abadi et al. (2016) provided stricter bounds on the privacy loss induced by a noisy SGD by introducing the moments accountant. In comparison with these efforts, our work increases the accuracy of a private MNIST model from $97\%$ to $98\%$ while improving the privacy bound $\varepsilon$ from 8 to 1.9. Furthermore, the PATE approach is independent of the learning algorithm, unlike this previous work. Support for a wide range of architecture and training algorithms allows us to obtain good privacy bounds on an accurate and private SVHN model. However, this comes at the cost of assuming that non-private unlabeled data is available, an assumption that is not shared by (Abadi et al., 2016; Shokri & Shmatikov, 2015).

Pathak et al. (2010) first discussed secure multi-party aggregation of locally trained classifiers for a global classifier hosted by a trusted third-party. Hamm et al. (2016) proposed the use of knowledge transfer between a collection of models trained on individual devices into a single model guaranteeing differential privacy. Their work studied linear student models with convex and continuously differentiable losses, bounded and $c$-Lipschitz derivatives, and bounded features. The PATE approach of this paper is not constrained to such applications, but is more generally applicable.

Previous work also studied semi-supervised knowledge transfer from private models. For instance, Jagannathan et al. (2013) learned privacy-preserving random forests. A key difference is that their approach is tailored to decision trees. PATE works well for the specific case of decision trees, as demonstrated in Appendix C, and is also applicable to other machine learning algorithms, including more complex ones. Another key difference is that Jagannathan et al. (2013) modified the classic model of a decision tree to include the Laplacian mechanism. Thus, the privacy guarantee does not come from the disjoint sets of training data analyzed by different decision trees in the random forest, but rather from the modified architecture. In contrast, partitioning is essential to the privacy guarantees of the PATE approach.

## 6    CONCLUSIONS

To protect the privacy of sensitive training data, this paper has advanced a learning strategy and a corresponding privacy analysis. The PATE approach is based on knowledge aggregation and transfer from "teacher" models, trained on disjoint data, to a "student" model whose attributes may be made public. In combination, the paper's techniques demonstrably achieve excellent utility on the MNIST and SVHN benchmark tasks, while simultaneously providing a formal, state-of-the-art bound on users' privacy loss. While our results are not without limits—e.g., they require disjoint training data for a large number of teachers (whose number is likely to increase for tasks with many output classes)—they are encouraging, and highlight the advantages of combining semi-supervised learning with precise, data-dependent privacy analysis, which will hopefully trigger further work. In particular, such future work may further investigate whether or not our semi-supervised approach will also reduce teacher queries for tasks other than MNIST and SVHN, for example when the discrete output categories are not as distinctly defined by the salient input space features.

A key advantage is that this paper's techniques establish a precise guarantee of training data privacy in a manner that is both intuitive and rigorous. Therefore, they can be appealing, and easily explained, to both an expert and non-expert audience. However, perhaps equally compelling are the techniques' wide applicability. Both our learning approach and our analysis methods are "black-box," i.e., independent of the learning algorithm for either teachers or students, and therefore apply, in general, to non-convex, deep learning, and other learning methods. Also, because our techniques do not constrain the selection or partitioning of training data, they apply when training data is naturally and non-randomly partitioned—e.g., because of privacy, regulatory, or competitive concerns—or when each teacher is trained in isolation, with a different method. We look forward to such further applications, for example on RNNs and other sequence-based models.

### ACKNOWLEDGMENTS

Nicolas Papernot is supported by a Google PhD Fellowship in Security. The authors would like to thank Ilya Mironov and Li Zhang for insightful discussions about early drafts of this document.

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

## A  MISSING DETAILS ON THE ANALYSIS

We provide missing proofs from Section 3.

**Theorem 3.** *Let $\mathcal{M}$ be $(2\gamma, 0)$-differentially private and $q \geq \Pr[\mathcal{M}(d) \neq o^*]$ for some outcome $o^*$. Let $l, \gamma \geq 0$ and $q < \frac{e^{2\gamma}-1}{e^{4\gamma}-1}$. Then for any aux and any neighbor $d'$ of $d$, $\mathcal{M}$ satisfies*

$$\alpha(l; \textit{aux}, d, d') \leq \log((1-q)\Big(\frac{1-q}{1-e^{2\gamma}q}\Big)^l + q\exp(2\gamma l)).$$

*Proof.* Since $\mathcal{M}$ is $2\gamma$-differentially private, for every outcome $o$, $\frac{Pr[M(d)=o]}{Pr[M(d')=o]} \leq \exp(2\gamma)$. Let $q' = Pr[M(d) \neq o^*]$. Then $Pr[M(d') \neq o^*] \leq \exp(2\gamma)q'$. Thus

$$\exp(\alpha(l; \textit{aux}, d, d')) = \sum_o \Pr[M(d) = o]\Big(\frac{\Pr[M(d) = o]}{\Pr[M(d') = o]}\Big)^l$$

$$= \Pr[M(d) = o^*]\Big(\frac{\Pr[M(d) = o^*]}{\Pr[M(d') = o^*]}\Big)^l + \sum_{o \neq o^*} \Pr[M(d) = o]\Big(\frac{\Pr[M(d) = o]}{\Pr[M(d') = o]}\Big)^l$$

$$\leq (1-q')\Big(\frac{1-q'}{1-e^{2\gamma}q'}\Big)^l + \sum_{o \neq o^*} \Pr[M(d) = o](e^{2\gamma})^l$$

$$\leq (1-q')\Big(\frac{1-q'}{1-e^{2\gamma}q'}\Big)^l + q'e^{2\gamma l}.$$

Now consider the function

$$f(z) = (1-z)\Big(\frac{1-z}{1-e^{2\gamma}z}\Big)^l + ze^{2\gamma l}.$$

We next argue that this function is non-decreasing in $(0, \frac{e^{2\gamma}-1}{e^{4\gamma}-1})$ under the conditions of the lemma. Towards this goal, define

$$g(z, w) = (1-z)\Big(\frac{1-w}{1-e^{2\gamma}w}\Big)^l + ze^{2\gamma l},$$

and observe that $f(z) = g(z, z)$. We can easily verify by differentiation that $g(z, w)$ is increasing individually in $z$ and in $w$ in the range of interest. This implies that $f(q') \leq f(q)$ completing the proof. $\square$

**Lemma 4.** *Let $\mathbf{n}$ be the label score vector for a database $d$ with $n_{j^*} \geq n_j$ for all $j$. Then*

$$\Pr[\mathcal{M}(d) \neq j^*] \leq \sum_{j \neq j^*} \frac{2 + \gamma(n_{j^*} - n_j)}{4\exp(\gamma(n_{j^*} - n_j))}$$

*Proof.* The probability that $n_{j^*} + Lap(\frac{1}{\gamma}) < n_j + Lap(\frac{1}{\gamma})$ is equal to the probability that the sum of two independent $Lap(1)$ random variables exceeds $\gamma(n_{j^*} - n_j)$. The sum of two independent $Lap(1)$ variables has the same distribution as the difference of two $Gamma(2, 1)$ random variables. Recalling that the $Gamma(2, 1)$ distribution has pdf $xe^{-x}$, we can compute the pdf of the difference via convolution as

$$\int_{y=0}^{\infty} (y + |x|)e^{-y-|x|}ye^{-y} \, dy = \frac{1}{e^{|x|}} \int_{y=0}^{\infty} (y^2 + y|x|)e^{-2y} \, dy = \frac{1 + |x|}{4e^{|x|}}.$$

The probability mass in the tail can then be computed by integration as $\frac{2+\gamma(n_{j^*}-n_j)}{4\exp(\gamma(n_{j^*}-n_j))}$. Taking a union bound over the various candidate $j$'s gives the claimed bound. $\square$

## B  APPENDIX: TRAINING THE STUDENT WITH MINIMAL TEACHER QUERIES

In this appendix, we describe approaches that were considered to reduce the number of queries made to the teacher ensemble by the student during its training. As pointed out in Sections 3 and 4, this effort is motivated by the direct impact of querying on the total privacy cost associated with student training. The first approach is based on *distillation*, a technique used for knowledge transfer and model compression (Hinton et al., 2015). The three other techniques considered were proposed in the context of *active learning*, with the intent of identifying training examples most useful for learning. In Sections 2 and 4, we described semi-supervised learning, which yielded the best results. The student models in this appendix differ from those in Sections 2 and 4, which were trained using GANs. In contrast, all students in this appendix were learned in a fully supervised fashion from a subset of public, labeled examples. Thus, the learning goal was to identify the subset of labels yielding the best learning performance.

### B.1  TRAINING STUDENTS USING DISTILLATION

Distillation is a knowledge transfer technique introduced as a means of compressing large models into smaller ones, while retaining their accuracy (Bucilua et al., 2006; Hinton et al., 2015). This is for instance useful to train models in data centers before deploying compressed variants in phones. The transfer is accomplished by training the smaller model on data that is labeled with probability vectors produced by the first model, which encode the knowledge extracted from training data. Distillation is parameterized by a *temperature* parameter $T$, which controls the smoothness of probabilities output by the larger model: when produced at small temperatures, the vectors are discrete, whereas at high temperature, all classes are assigned non-negligible values. Distillation is a natural candidate to compress the knowledge acquired by the ensemble of teachers, acting as the large model, into a student, which is much smaller with $n$ times less trainable parameters compared to the $n$ teachers.

To evaluate the applicability of distillation, we consider the ensemble of $n = 50$ teachers for SVHN. In this experiment, we do not add noise to the vote counts when aggregating the teacher predictions. We compare the accuracy of three student models: the first is a baseline trained with labels obtained by plurality, the second and third are trained with distillation at $T \in \{1, 5\}$. We use the first 10,000 samples from the test set as unlabeled data. Figure 5 reports the accuracy of the student model on the last 16,032 samples from the test set, which were not accessible to the model during training. It is plotted with respect to the number of samples used to train the student (and hence the number of queries made to the teacher ensemble). Although applying distillation yields classifiers that perform more accurately, the increase in accuracy is too limited to justify the increased privacy cost of revealing the entire probability vector output by the ensemble instead of simply the class assigned the largest number of votes. Thus, we turn to an investigation of active learning.

### B.2  ACTIVE LEARNING OF THE STUDENT

Active learning is a class of techniques that aims to identify and prioritize points in the student's training set that have a high potential to contribute to learning (Angluin, 1988; Baum, 1991). If the label of an input in the student's training set can be predicted confidently from what we have learned so far by querying the teachers, it is intuitive that querying it is not worth the privacy budget spent. In our experiments, we made several attempts before converging to a simpler final formulation.

**Siamese networks:** Our first attempt was to train a pair of siamese networks, introduced by Bromley et al. (1993) in the context of one-shot learning and later improved by Koch (2015). The siamese networks take two images as input and return 1 if the images are equal and 0 otherwise. They are two identical networks trained with shared parameters to force them to produce similar representations of the inputs, which are then compared using a distance metric to determine if the images are identical or not. Once the siamese models are trained, we feed them a pair of images where the first is unlabeled and the second labeled. If the unlabeled image is confidently matched with a known labeled image, we can infer the class of the unknown image from the labeled image. In our experiments, the siamese networks were able to say whether two images are identical or not, but did not generalize well: two images of the same class did not receive sufficiently confident matches. We also tried a variant of this approach where we trained the siamese networks to output 1 when the two

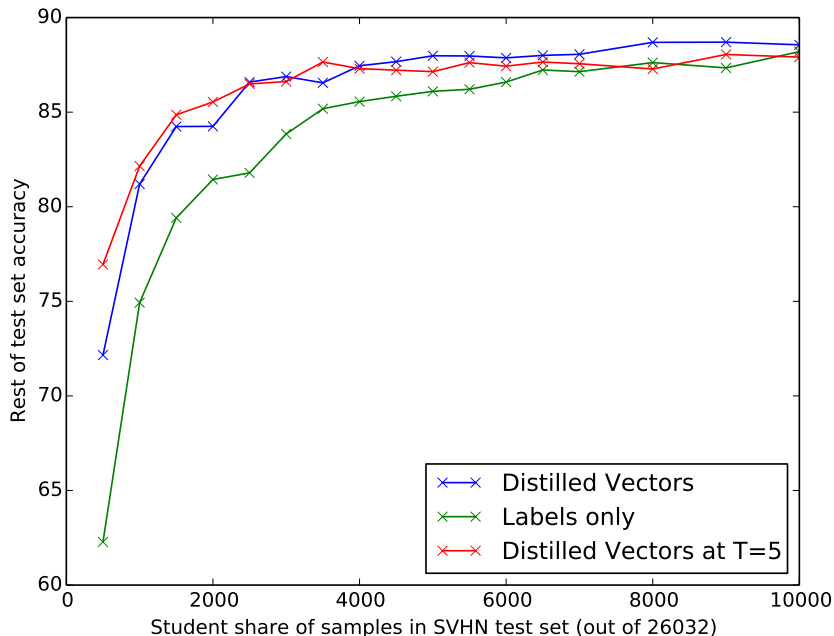

Figure 5: Influence of distillation on the accuracy of the SVHN student trained with respect to the initial number of training samples available to the student. The student is learning from $n = 50$ teachers, whose predictions are aggregated without noise: in case where only the label is returned, we use plurality, and in case a probability vector is returned, we sum the probability vectors output by each teacher before normalizing the resulting vector.

images are of the same class and $0$ otherwise, but the learning task proved too complicated to be an effective means for reducing the number of queries made to teachers.

**Collection of binary experts:** Our second attempt was to train a collection of binary experts, one per class. An expert for class $j$ is trained to output $1$ if the sample is in class $j$ and $0$ otherwise. We first trained the binary experts by making an initial batch of queries to the teachers. Using the experts, we then selected available unlabeled student training points that had a candidate label score below $0.9$ and at least $4$ other experts assigning a score above $0.1$. This gave us about $500$ unconfident points for $1700$ initial label queries. After labeling these unconfident points using the ensemble of teachers, we trained the student. Using binary experts improved the student's accuracy when compared to the student trained on arbitrary data with the same number of teacher queries. The absolute increases in accuracy were however too limited—between $1.5\%$ and $2.5\%$.

**Identifying unconfident points using the student:** This last attempt was the simplest yet the most effective. Instead of using binary experts to identify student training points that should be labeled by the teachers, we used the student itself. We asked the student to make predictions on each unlabeled training point available. We then sorted these samples by increasing values of the maximum probability assigned to a class for each sample. We queried the teachers to label these unconfident inputs first and trained the student again on this larger labeled training set. This improved the accuracy of the student when compared to the student trained on arbitrary data. For the same number of teacher queries, the absolute increases in accuracy of the student trained on unconfident inputs first when compared to the student trained on arbitrary data were in the order of $4\% - 10\%$.

## C    APPENDIX: ADDITIONAL EXPERIMENTS ON THE UCI ADULT AND DIABETES DATASETS

In order to further demonstrate the general applicability of our approach, we performed experiments on two additional datasets. While our experiments on MNIST and SVHN in Section 4 used convolutional neural networks and GANs, here we use random forests to train our teacher and student models for both of the datasets. Our new results on these datasets show that, despite the differing data types and architectures, we are able to provide meaningful privacy guarantees.

**UCI Adult dataset:** The UCI Adult dataset is made up of census data, and the task is to predict when individuals make over $50k per year. Each input consists of 13 features (which include the age, workplace, education, occupation—see the UCI website for a full list[3]). The only pre-processing we apply to these features is to map all categorical features to numerical values by assigning an integer value to each possible category. The model is a random forest provided by the `scikit-learn` Python package. When training both our teachers and student, we keep all the default parameter values, except for the number of estimators, which we set to 100. The data is split between a training set of 32,562 examples, and a test set of 16,282 inputs.

**UCI Diabetes dataset:** The UCI Diabetes dataset includes de-identified records of diabetic patients and corresponding hospital outcomes, which we use to predict whether diabetic patients were readmitted less than 30 days after their hospital release. To the best of our knowledge, no particular classification task is considered to be a standard benchmark for this dataset. Even so, it is valuable to consider whether our approach is applicable to the likely classification tasks, such as readmission, since this dataset is collected in a medical environment—a setting where privacy concerns arise frequently. We select a subset of 18 input features from the 55 available in the dataset (to avoid features with missing values) and form a dataset balanced between the two output classes (see the UCI website for more details[4]). In class 0, we include all patients that were readmitted in a 30-day window, while class 1 includes all patients that were readmitted after 30 days or never readmitted at all. Our balanced dataset contains 34,104 training samples and 12,702 evaluation samples. We use a random forest model identical to the one described above in the presentation of the Adult dataset.

**Experimental results:** We apply our approach described in Section 2. For both datasets, we train ensembles of $n = 250$ random forests on partitions of the training data. We then use the noisy aggregation mechanism, where vote counts are perturbed with Laplacian noise of scale 0.05 to privately label the first 500 test set inputs. We train the student random forest on these 500 test set inputs and evaluate it on the last 11,282 test set inputs for the Adult dataset, and 6,352 test set inputs for the Diabetes dataset. These numbers deliberately leave out some of the test set, which allowed us to observe how the student performance-privacy trade-off was impacted by varying the number of private labels, as well as the Laplacian scale used when computing these labels.

For the Adult dataset, we find that our student model achieves an 83% accuracy for an $(\varepsilon, \delta) = (2.66, 10^{-5})$ differential privacy bound. Our non-private model on the dataset achieves 85% accuracy, which is comparable to the state-of-the-art accuracy of 86% on this dataset (Poulos & Valle, 2016). For the Diabetes dataset, we find that our privacy-preserving student model achieves a 93.94% accuracy for a $(\varepsilon, \delta) = (1.44, 10^{-5})$ differential privacy bound. Our non-private model on the dataset achieves 93.81% accuracy.

---

[3] https://archive.ics.uci.edu/ml/datasets/Adult
[4] https://archive.ics.uci.edu/ml/datasets/Diabetes+130-US+hospitals+for+years+1999-2008

