# Peer review of "Semi-supervised Knowledge Transfer for Deep Learning from Private Training Data"

_ICLR 2017 — accepted_

[Public Comment · (anonymous) · 14 Dec 2016]
**Input of student model**

Thanks for interesting and well-organized papers. I have a question about teacher-student model. 

Teachers are trained on sensitive data, and students are trained on non-sensitive data.
I wonder how students work on the outputs of teachers.
Sensitive and non-sensitive are different attributes, so I think there are no correlation between teachers and students. 

Please give me some more details. Thanks.

[Official Review · AnonReviewer3 · rating 7 · confidence 3 · 16 Dec 2016]
**Good theory**
originality 5 · clarity 5

This paper discusses how to guarantee privacy for training data. In the proposed approach multiple models trained with disjoint datasets are used as ``teachers'' model, which will train a ``student'' model to predict an output chosen by noisy voting among all of the teachers. 

The theoretical results are nice but also intuitive. Since teachers' result are provided via noisy voting, the student model may not duplicate the teacher's behavior. However, the probabilistic bound has quite a number of  empirical parameters, which makes me difficult to decide whether the security is 100% guaranteed or not.

The experiments on MNIST and SVHN are good. However, as the paper claims, the proposed approach may be mostly useful for sensitive data like medical histories, it will be nice to conduct one or two experiments on such applications.

[Official Review · AnonReviewer2 · rating 9 · confidence 4 · 16 Dec 2016]
**A nice contribution to differentially-private deep learning**
originality 5 · clarity 5 · impact 5 · appropriateness 5 · recommendation (unofficial) 5

Altogether a very good paper, a nice read, and interesting. The work advances the state of the art on differentially-private deep learning, is quite well-written, and relatively thorough.

One caveat is that although the approach is intended to be general, no theoretical guarantees are provided about the learning performance. Privacy-preserving machine learning papers often analyze both the privacy (in the worst case, DP setting) and the learning performance (often under different assumptions). Since the learning performance might depend on the choice of architecture; future experimentation is encouraged, even using the same data sets, with different architectures. If this will not be added, then please justify the choice of architecture used, and/or clarify what can be generalized about the observed learning performance.

Another caveat is that the reported epsilons are not those that can be privately released; the authors note that their technique for doing so would change the resulting epsilon. However this would need to be resolved in order to have a meaningful comparison to the epsilon-delta values reported in related work.

Finally, as has been acknowledged in the paper, the present approach may not work on other natural data types. Experiments on other data sets is strongly encouraged. Also, please cite the data sets used.

Other comments:

Discussion of certain parts of the related work are thorough. However, please add some survey/discussion of the related work on differentially-private semi-supervised learning. For example, in the context of random forests, the following paper also proposed differentially-private semi-supervised learning via a teacher-learner approach (although not denoted as “teacher-learner”). The only time the private labeled data is used is when learning the “primary ensemble.”  A "secondary ensemble" is then learned only from the unlabeled (non-private) data, with pseudo-labels generated by the primary ensemble.

G. Jagannathan, C. Monteleoni, and K. Pillaipakkamnatt: A Semi-Supervised Learning Approach to Differential Privacy. Proc. 2013 IEEE International Conference on Data Mining Workshops, IEEE Workshop on Privacy Aspects of Data Mining (PADM), 2013.

Section C. does a nice comparison of approaches. Please make sure the quantitative results here constitute an apples-to-apples comparison with the GAN results. 

The paper is extremely well-written, for the most part. Some places needing clarification include:
- Last paragraph of 3.1. “all teachers….get the same training data….” This should be rephrased to make it clear that it is not the same w.r.t. all the teachers, but w.r.t. the same teacher on the neighboring database.
- 4.1: The authors state: “The number n of teachers is limited by a trade-off between the classification task’s complexity and the available data.” However, since this tradeoff is not formalized, the statement is imprecise. In particular, if the analysis is done in the i.i.d. setting, the tradeoff would also likely depend on the relation of the target hypothesis to the data distribution.
- Discussion of figure 3 was rather unclear in the text and caption and should be revised for clarity. In the text section, at first the explanation seems to imply that a larger gap is better (as is also indicated in the caption). However later it is stated that the gap stays under 20%. These sentences seem contradictory, which is likely not what was intended.

[Official Review · AnonReviewer1 · rating 9 · confidence 4 · 17 Dec 2016]
**Nice paper, strong accept**
originality 5 · clarity 5 · impact 4

This paper addresses the problem of achieving differential privacy in a very general scenario where a set of teachers is trained on disjoint subsets of sensitive data and the student performs prediction based on public data labeled by teachers through noisy voting. I found the approach altogether plausible and very clearly explained by the authors. Adding more discussion of the bound (and its tightness) from Theorem 1 itself would be appreciated. A simple idea of adding perturbation error to the counts, known from differentially-private literature, is nicely re-used by the authors and elegantly applied in a much broader (non-convex setting) and practical context than in a number of differentially-private and other related papers. The generality of the approach, clear improvement over predecessors, and clarity of the writing makes the method worth publishing.

[Public Comment · (anonymous) · 10 Jan 2017]
**Attacker's Model and Goal?**

Hi,

I have few questions about the paper.

1- What attacker's goal did you consider in your paper? Is it recovering the training data, or checking whether a specific sample has been in the training data? 

2- If the attacker's goal is to recover the training data, does the attacker want to recover the exact data or an approximation would be OK?

3- Talking about neural networks:
- Do you think there is any attack method to recover an exact training data from the learning model?
- Do you think there is any defense method to prevent an attacker from recovering even an approximate training data?

4- How can we quantify the strength of a learning model (specifically neural networks) without any defensive mechanism?

5- How can we quantify the strength of a learning model which has not been trained on exact training data? For example, some forms of adversarial training methods never train the model on the clean data; instead, at each epoch, the model is trained on different adversarial data derived from the real data. 
- How can the model "memorize" the training data, when 1) it has never seen the real data, 2) it has been trained on different data in different epochs?

6- How do you compare the performance of your method with adversarial training?

Thanks.

[Public Comment · (anonymous) · 11 Jan 2017 (modified: 12 Jan 2017)]
**Question for student GAN training**

Thank you for providing an interesting paper.

In the paper, the student model is trained by semi-supervised fashion as suggested in (Salimans et al., 2016).

As far as I understand, teacher's ensembles are using for supervised learning and nonsensitive data is for unsupervised learning.

So, my question is "Where is the generator ?".

The aggregation of teacher network is treated as the generator in GAN framework?

[Public Comment · (anonymous) · 12 Jan 2017]
**Simple question related with privacy loss**

I think this paper has great impact.

My question is what is the "auxiliary input" in Definition 2.

Could you explain this term in theoretical view and what is that in your paper?

[Final Decision · Program Chairs · 06 Feb 2017]
**ICLR committee final decision**

The paper presents a general teacher-student approach for differentially-private learning in which the student learns to predict a noise vote among a set of teachers. The noise allows the student to be differentially private, whilst maintaining good classification accuracies on MNIST and SVHN. The paper is well-written.